# Proteomic Analysis to Understand the Promotive Effect of Ethanol on Soybean Growth Under Salt Stress

**DOI:** 10.3390/biology13110861

**Published:** 2024-10-24

**Authors:** Setsuko Komatsu, Takumi Nishiuchi

**Affiliations:** 1Department of Applied Chemistry and Food Science, Fukui University of Technology, Fukui 910-8505, Japan; 2Division of Integrated Omics Research, Bioscience Core Facility, Research Center for Experimental Modeling of Human Disease, Kanazawa University, Kanazawa 920-8640, Japan; tnish9@staff.kanazawa-u.ac.jp

**Keywords:** proteomics, soybean, salt stress, ethanol

## Abstract

Soybean is a globally important resource crop of oil and protein but is sensitive to salinity. Soybean growth was significantly reduced under salt stress; however, it was restored and comparable to control values by ethanol application even under stress. To study the positive mechanism of ethanol on soybean growth, a proteomic approach was carried out. The categories with the greatest changes in protein numbers were protein metabolism, transport, and cell organization in biological processes. Reactive oxygen species enzymes were increased under salt stress; among them, mitochondrial ascorbate peroxidase was further accumulated by ethanol application. Among the cell wall and membrane-associated proteins, xyloglucan xyloglucosyl transferase and H^+^-ATPase increased and decreased, respectively, under salt stress; however, they were restored to control levels by ethanol application. These results suggest that soybeans were adversely affected by salt stress and recovered with ethanol application via the regulation of cell wall and membrane functions through the detoxification of reactive oxygen species.

## 1. Introduction

The soybean (*Glycine max*) is a globally important resource crop of oil and protein but is sensitive to salinity with yield losses of up to 40% [1]. Stress induced by salt negatively regulated gibberellic acid biogenesis and positively mediated abscisic acid biogenesis, delaying soybean seed germination [2]. Salt stress in soybean leaves induced dynamic lipid alterations in the recycling of both phospholipids and galactolipids [3]. Proteins related to the phagosome, spliceosome, and soluble *N*-ethylmaleimide-sensitive-factor attachment receptor accumulated in leaves, while proteins related to fatty-acid biosynthesis, linoleic acid metabolism, and endocytosis were abundant in the root of salt-stressed soybeans [4]. High salinity induced the accumulation of proteins with binding/catalytic activity in soybean roots [5]. DNA methylation was a fundamental event for initiating a transcriptional response in salt-treated soybeans [6]. These findings indicated that a comprehensive recognition of soybean responses to salt stress at molecular and physiological levels is important for conferring stress tolerance in soybeans.

High sodium ion concentrations in the soil increase osmotic pressure, disrupt cellular ion homeostasis, and inhibit water/nutrient absorption, which negatively impacts soybean growth and reduces agricultural yields [7]. Salt stress induces the accumulation of reactive oxygen species (ROS), which act as secondary stress factors and induce membrane lipid peroxidation, leading to the degrading of the structure of cell membrane proteins [8]. Furthermore, the interactions among ribosomal metabolism, mitogen-activated protein kinase signaling, phenylpropanoid biosynthesis, starch/sucrose metabolism, and plant hormone signaling play important functions in the response of soybeans to salt stress [9]. In addition, the application of plant-derived smoke solution enhanced soybean growth by alleviating the effect of salt stress by controlling energy metabolism, protein glycosylation, and cell wall assembly [10]. Moreover, the application of safranal to soybeans improved salt tolerance by regulating the cell wall along with controlling the ROS scavenging system [11]. These findings indicated that the modification of multiple intracellular systems could confer salt stress tolerance in soybeans.

Chemical priming, which allows for the flexible activation of adaptive control responses to adverse conditions using safe agents, is a complementary method to improve stress tolerance. During stress responses in plants, ethanol fermentation is one of the vital processes and is required for the response to hypoxic stress [12]. Under anaerobic stress as a part of the fermentation pathway in plants, endogenous ethanol is formed [13]. Ethanol had a positive regulatory role in improving the growth of plants such as the tomato (*Solanum lycopersicum*) [14] and strawberry (*Fragaria ananassa*) [15]. On the other hand, the stress tolerance against high salinity in *Arabidopsis thaliana* induced hypoxic conditions, leading to ethanol fermentation [16]. Ethanol was identified to enhance chilling tolerance in rice [17], drought tolerance in soybeans [18], and heat stress tolerance in lettuce (*Lactuca sativa* L.) [19], as well as salt tolerance in *Arabidopsis*, rice (*Oryza sativa*), and soybeans [20,21]. In *Arabidopsis* and rice, a high salinity tolerance caused by ethanol application upregulated the expression of ROS signaling-related genes under salt stress [20,22]. Ethanol might have an important role in mitigating the negative effects of salt on economically crucial crops. However, it is not fully elucidated how ethanol treatment can enhance salt-stress tolerance in crops such as soybeans.

In soybeans, ethanol increased the contents of total soluble sugars and free amino acids, as well as the ROS-scavenging enzyme in leaves under drought stress, implying that ethanol employed these compounds for osmotic adjustment in plants under stress [18]. Furthermore, ethanol enhanced biochemical and physiological responses to alleviate saline toxicity by suppressing oxidative stress and subsequent cellular damage by limiting the excessive accumulation of ROS [21]. Ethanol’s potential should be further investigated as an antioxidant and growth promoter for soybeans. The objective of this study is to clarify the ethanol-induced tolerance to salt stress during the early growth stages of soybeans from the perspective of protein science. In the beginning, to investigate the promoting effect of ethanol on soybeans, morphological measurements were carried out. Following this result, nano-liquid chromatography (LC) combined with mass spectrometry (MS) analysis was conducted to explore the underlying mechanisms in cells. To validate the proteomic results, immunoblot analysis was performed.

## 2. Materials and Methods

### 2.1. Plant Material and Treatment

Soybean (*Glycine max* L. cultivar Enrei) seeds were sown in quartz sand in a nursery case. Three-day-old seedlings were treated with or without 300 mM of ethanol (Nacalai Tesque, Kyoto, Japan) as well as with or without 150 mM of NaCl (Nacalai Tesque) for 2 days. For treatment, ethanol and NaCl were added as 100 mL of solution to the sand and absorbed through the roots. Seedlings were kept at 25 °C in a growth chamber illuminated with white fluorescent lights (200 μmol m^−2^ s^−1^, 12 h light period/day). For morphological, proteomic, and conformational analyses, hypocotyls and roots from 5-day-old seedlings were collected (Figure 1). For all experiments, 3 independent experiments were performed, which were sown on different days as independent biological replicates.

### 2.2. Protein Extraction

Samples (500 mg) were ground with a mortar and pestle in 500 μL of extraction buffer containing 50 mM of Tris-HCl (pH 7.6), 100 mM of NaCl, 1% Nonidet-P40, 0.1% sodium dodecyl sulfate (SDS), and protease inhibitors (Nacalai Tesque). The suspension was centrifuged twice with 16,000× *g* for 10 min at 4 °C, and the supernatant was used as the soluble fraction. Protein concentration was measured at 595 nm using Bradford methods [23] with bovine serum albumin as the standard.

### 2.3. Protein Enrichment, Reduction, Alkylation, Digestion, and Desalting

Quantified proteins (100 μg) were adjusted to a final volume of 100 μL, to which 400 μL of methanol was added and mixed, followed by the addition of 100 μL of chloroform and 300 μL of water. After centrifugation at 16,000× *g* for 10 min, the solution was discarded, and 300 μL of methanol was added. After centrifugation at 16,000× *g* for 10 min, the pellet was collected as the soluble fraction [24]. Proteins were resuspended in 50 mM of ammonium bicarbonate, reduced with 50 mM of dithiothreitol for 30 min at 56 °C, and alkylated with 50 mM of iodoacetamide for 30 min at 37 °C in the dark. Alkylated proteins were digested with trypsin (FUJIFILM Wako Chemical, Osaka, Japan) at an enzyme/protein mass ratio of 1:100 for 18 h at 37 °C. Peptides were desalted on a MonoSpin C18 Column (GL Sciences, Tokyo, Japan) and acidified with 1% trifluoroacetic acid.

### 2.4. Protein Identification Using nanoLC-MS/MS

The LC conditions, as well as the MS acquisition conditions, are described in the previous study [25]. The peptides were loaded onto the LC system (EASY-nLC 1200; Thermo Fisher Scientific, San Jose, CA, USA), equilibrated with 0.1% formic acid, and eluted with a linear acetonitrile gradient (0–35%) in 0.1% formic acid at a flow rate of 300 nL min^−1^. The eluted peptides were loaded and separated on the Aurora column (25 cm × 75 μm ID, 1.6 mm C18; IonOpticks, Fitzroy, Austria) with a spray voltage of 1.5 kV (Ion Transfer Tube temperature: 275 °C). The peptide ions were detected using MS (Orbitrap Fusion ETD MS; Thermo Fisher Scientific) in the data-dependent acquisition mode with the installed Xcalibur software (version 4.0; Thermo Fisher Scientific). Full-scan mass spectra were acquired in the MS over 375–1500 *m*/*z* with a resolution of 70,000. The most intense precursor ions were selected for collision-induced fragmentation in the linear ion trap at a normalized collision energy of 35%. Dynamic exclusion was employed within 15 s to prevent the repetitive selection of peptides.

### 2.5. MS-Data Analysis

The MS/MS searches were carried out using SEQUEST HT search algorithms against the UniprotKB *Glycine max* (Soybean) protein database (29 October 2022) using Proteome Discoverer 2.5 (Version 2.5.0.400; Thermo Fisher Scientific). Label-free quantification was also performed with Proteome Discoverer 2.5 using precursor ions detector nodes. The processing workflow included spectrum files RC, spectrum selector, SEQUEST HT search nodes, percolator, ptmRS, and minor feature detector nodes. The oxidation of methionine was set as a variable modification, and the carbamido-methylation of cysteine was set as a fixed modification. Mass tolerances in MS and MS/MS were set at 10 ppm and 0.6 Da, respectively. Trypsin was specified as protease, and a maximum of 2 missed cleavages was allowed. Target-decoy database searches were used to calculate the false discovery rate, and peptide identification was set at 1%.

### 2.6. Differential Analysis of Proteins Using MS Data

The consensus workflow included MSF files, Feature Mapper, precursor ion quantifier, PSM groper, peptide validator, peptide and protein filter, protein scorer, protein marker, protein false discovery rate validator, protein grouping, and peptide in protein. Normalization of the abundances was performed using the total peptide amount mode. Significance was assessed using the Abundance Ratio Adjusted *p*-value. Principal-component analysis was performed with Proteome Discoverer 2.5.

### 2.7. Immunoblot Analysis

Quantified proteins (10 μg) were added to an SDS sample buffer containing 60 mM of Tris-HCl (pH 6.8), 2% SDS, 5% dithiothreitol, 10% glycerol, and 0.01% bromophenol blue (Bio-Rad, Hercules, CA, USA) [26]. Proteins were separated by electrophoresis on a 10% SDS polyacrylamide gel. Coomassie-brilliant blue staining was used as a loading control. Proteins in the gel were transferred to a polyvinylidene difluoride (PVDF) membrane using a semi-dry transfer blotter. The PVDF membrane was blocked in Bullet Blocking One reagent (Nacalai Tesque) for 5 min and cross-reacted with a 1:1000 dilution of primary antibodies for 30 min. As primary antibodies, anti-ascorbate peroxidase [25], peroxiredoxin [27], xyloglucan xyloglucosyl transferase (Agrisera, Vannas, Sweden), cellulose synthetase (Agrisera), and H^+^-ATPase (Agrisera) antibodies were used. Anti-rabbit IgG conjugated with horseradish peroxidase (Bio-Rad) was used as the secondary antibody. After 30 min of incubation, signals were detected using a TMB Membrane Peroxidase Substrate Kit (Seracare, Milford, MA, USA). The integrated densities of bands were calculated using ImageJ software (version 1.53e with Java 1.8.0_172; National Institutes of Health, Bethesda, MD, USA).

### 2.8. Statistical Analysis

Statistical significance for 2 groups was assessed using the Student’s *t*-test. Statistical significance for multiple groups was assessed using a one-way ANOVA test. SPSS 20.0 (IBM, Chicago, IL, USA) statistical software was used to evaluate the results. A *p*-value of less than 0.05 was considered statistically significant.

## 3. Results

### 3.1. Morphological Changes in Soybean Seedlings Treated with Ethanol Under Salt Stress

To investigate the effect of ethanol on soybeans under salt stress, morphological analysis was performed. Three-day-old seedings were treated with or without 300 mM of ethanol and with or without 150 mM of NaCl for 2 days (Figure 1 and Appendix A). As morphological parameters, the length of the hypocotyl, fresh weight of the hypocotyl, length of the main root, and fresh weight of the total root were measured (Figure 2). The length of the hypocotyl, length of the main root, and fresh weight of the total root significantly decreased under salt stress. By applying ethanol, these three parameters were restored and comparable to control values even under salt stress. The fresh weight of the hypocotyl was suppressed by salt stress; however, it was not restored by ethanol application under stress (Figure 2). Based on the morphological results, roots were used for proteomic analysis.

### 3.2. Identification and Functional Classification of Soybean Root Proteins Altered by Ethanol Treatment Under Salt Stress

To clarify the subcellular mechanisms in the growth of soybeans treated with ethanol under salt stress, proteomics was performed (Appendix A). Four treatments were conducted: with or without ethanol and with or without salt stress (Figure 1). After treatment, proteins extracted from soybean roots were concentrated, reduced, alkylated, digested, and desalted. These steps were followed by an LC-MS/MS analysis, and a total of 8252 proteins were detected. After the LC-MS/MS analysis, the relative abundances of proteins in salt-stressed soybeans compared with control (Appendix A) and salt-stressed soybeans with ethanol application compared with salt stress only (Appendix A) were analyzed. Principal-component analysis was performed in Proteome Discoverer using proteins from six different kinds of samples from salt/control and salt + ethanol/salt, which indicated the differential accumulation pattern of proteins from four different kinds of treatments (Figure 3). This result identified that salt stress caused root proteins to separate into significantly different groups, whereas ethanol application brought the two groups closer together (Figure 3).

In soybean root under salt stress compared to control, 214 proteins were differentially altered in abundance with a *p*-value of ≤0.05, peptide number of ≥2, and fold change of >1.5 and/or 0.67. Of the 214 proteins, 173 and 41 proteins increased and decreased, respectively, under salt stress compared to the control condition (Appendix A). On the other hand, in soybean roots treated with ethanol under salt stress compared to stress only, 363 proteins differentially altered in abundance with a *p*-value of ≤0.05, peptide number of ≥2, and fold change of >1.5 and/or 0.67. Of the 363 proteins, 212 and 151 proteins increased and decreased, respectively, with ethanol treatment under salt stress compared to stress only (Appendix A).

The functional categories of the identified proteins were obtained using gene ontology analysis (Figure 4). The categories with the most change in protein numbers were protein metabolism, transport, and cell organization in biological processes, nucleus and cytosol in cellular components, and nucleic acid binding activity in molecular functions (Figure 4). To validate the results identified using proteomic analysis, significantly increased or decreased proteins were further analyzed using immunoblotting.

### 3.3. Immuno-Blot Analysis of Proteins Identified by Proteomics in Soybeans Treated with Ethanol Under Salt Stress

To better reveal the protein changes due to different treatments, immunoblot analysis was performed based on the proteomic results. Proteins extracted from the roots and hypocotyls of soybeans were separated using electrophoresis on an SDS-polyacrylamide gel, transferred to PVDF membranes, and cross-reacted with primary antibodies. A staining pattern with Coomassie-brilliant blue was used as a loading control (Appendix A). The integrated density of bands was measured using ImageJ software and calculated from the results of triplicated immunoblots (Appendix A).

Based on the proteomic analysis, ethanol application significantly increased and caused the further accumulation of ROS scavenging-related enzymes; this result was confirmed by immunoblot analysis. Because proteins belonging to the ascorbate peroxidase family, such as I1MZT0/I1MZT4/C6TB83 (Appendix A) and C6T7D4/I1KEL7 (Appendix A), among the ROS scavenging-related enzymes were altered in the proteomic data, the abundance of ascorbate peroxidase and peroxiredoxin was selectively analyzed using immunoblot analysis (Figure 5, Appendix A). Peroxiredoxin accumulated in hypocotyls in response to salt stress but was not altered by ethanol application (Figure 5C). When ethanol was applied under salt stress, cytoplasmic ascorbate peroxidase increased in hypocotyls compared to stress alone (Figure 5A). The abundance of mitochondrial ascorbate peroxidase increased with salt stress and was further enhanced by ethanol application under stress (Figure 5B).

Based on proteomic analysis, because cell wall and membrane-associated proteins were significantly increased and restored by ethanol application, this result was confirmed using immunoblot analysis (Figure 6). In the proteomic results, cellulose synthetase, plasma membrane ATPase, and xyloglucan endotransglucosylase/hydrolase were significantly altered (Appendix A); especially, cellulose synthetase I1LU34 (Appendix A) increased with salt stress and decreased with ethanol application under stress (Appendix A). In the immunoblot results, xyloglucan xyloglucosyl transferase and cellulose synthetase accumulated in the roots with salt treatment and were restored to the control level by ethanol treatment, even with salt stress (Figure 6A,B). On the other hand, the accumulation of H^+^-ATPase in the roots was decreased under salt stress but was recovered to the control level by ethanol application, even under stress (Figure 6C).

## 4. Discussion

Salt stress adversely affects the development and growth of soybeans [28]. Specifically, the length of the shoot/root and the dry weight of the leaf/stem/root in soybeans significantly decreased with increasing salt stress levels compared to the control [29]. In the present study, soybean growth was significantly reduced by salt stress (Figure 2). These results and previous studies generally suggest that salt stress has a detrimental effect on soybeans. The development of techniques to improve plant tolerance to high-salinity stress is important, and many approaches have been reported [30]. Among these approaches, ethanol played a positive regulatory role in improving the growth performance of plants such as tomatoes [14] and strawberries [15]. These demonstrate that various kinds of crop growth can be promoted by ethanol treatment alone.

Ethanol caused improvement in salt-stress tolerance in *Arabidopsis* [20], rice [22], and soybeans [21]. The exposure of 12-day-old soybean plants to salt stress for 7 days resulted in the distortion of plant morphological traits. Conversely, ethanol application to the leaves of salt-stressed plants reduced canopy wilting and yellowing, as well as recovered growth rate and biomass production [21]. In this study, soybean roots at the early growth stage were used to compare the effect with its vegetative stage. The negative phenotype of soybeans by salt stress was restored and comparable to control values when ethanol was applied under stress (Figure 2). These results, together with previous findings, indicate that soybean is sensitive to salt stress regardless of the growth stage, but the addition of ethanol might increase its tolerance to stress.

Increasing the salt concentration in soil disrupts the osmotic balance, causing water deficiency, ion toxicity, and oxidative stress in plants [31]. Plants evolved potent antioxidant defense mechanisms, which include enzymatic and nonenzymatic antioxidants, to counter ROS-induced oxidative injury under salt stress [32]. Various ROS scavenging enzymes were involved in reducing excess ROS generated under stress conditions [33]. *Ascorbate peroxidase* in ethanol-treated *Arabidopsis* was more unregulated than in untreated plants under salt stress; additionally, cytosolic ascorbate peroxidase was activated by ethanol in response to stress [20]. The ethanol-supplemented soybeans under salt stress further enhanced the activities of ascorbate peroxidase, peroxidase, glutathione S-transferase, and catalase, compared with salt-stressed plants alone [21]. In this study, mitochondrial ascorbate peroxidase, among ROS scavenging enzymes, was further accumulated by ethanol application compared to salt stress (Figure 5). These results, with previous knowledge, imply that ethanol improves salt-stress tolerance by detoxifying ROS.

Salt stress has a pronounced effect on the ultrastructure of plant cells, including responses such as abnormal chloroplast morphology, mitochondrial swelling, and cell wall thickening [34,35]. ROS are toxic to proteins, lipids, carbohydrates, and DNA, leading to membrane damage and cell death [36]. Excess hydroxyl radicals react with lipids and cause the degradation of cell membranes, which is a key barrier for protecting plant cells [37]. Scavenging toxic ROS during salt stress played a pinnacle role in protecting the cell membrane from oxidative damage [38,39]. Ascorbate peroxidase can neutralize hydrogen peroxide, protect cells from free radicals and oxidizing substances, as well as control the normal function of cells/organisms [40]. Among cell wall/cell membrane-related proteins, xyloglucan xyloglucosyl transferase and cellulose synthase, which increased under salt stress, were recovered to the control level by ethanol application (Figure 6). These results, together with previous studies, suggest that ROS generated by salt stress damages the cell membrane and cell wall, but ethanol may alleviate the effects by enhancing ROS scavenging mechanisms.

Among proteins related to the plasma membrane, the H^+^-ATPase decreased by salt recovered to control levels by ethanol application (Figure 6). Elevated salinity impedes the flow of electrons from the central transport chain to the oxygen-reduction pathways in various organelles, leading to excessive ROS generation in plants [41], with adverse effects on cell expansion [42], photosynthesis [43], and ion homeostasis [44]. The elevating H^+^-ATPase activity is necessary to form the H^+^ gradient to activate the Na^+^/H^+^ antiporter and eliminate excess Na^+^ to enhance salt tolerance [45]. In the case of boron application, the ameliorative effect was based on H^+^-ATPase stimulation and subsequent K^+^ retention via auxin- and ROS-mediated pathways [46]. Furthermore, the overexpression of rapeseed H^+^-ATPase (BnHA9) in *Arabidopsis* improved salt tolerance in transgenic plants [47]. These previous findings and the present results suggest that ethanol rescues soybeans from salt stress by enhancing H^+^-ATPase.

Various approaches, including biotechnological strategies, cultivation/breeding, and gene discovery, have been implemented to improve soybean salt tolerance [30]. However, because these approaches require more investment and time to develop stress-tolerant crops, farmers prefer simple and cheap approaches, which provide immediate agricultural and economic benefits [48]. For these reasons, ethanol is considered an excellent representative of an organic and cost-effective molecule.

## 5. Conclusions

Because soil salinity significantly reduces plant growth and grain yield of soybeans, finding solutions to reduce the effects of stress on soybeans is important for food security. Meanwhile, ethanol plays an important role in reducing the adverse effects of salinity on crops. In this study, soybean roots were significantly reduced under salt stress, but they were restored and comparable to control values by ethanol application even under stress. Proteomic and immunoblotting techniques were used to study the mechanism of the positive effect of ethanol on soybean growth. Key findings included the following: (i) The categories with the greatest changes in protein numbers were protein metabolism, transport, and cell organization in biological processes, nucleus and cytosol in cellular components, and nucleic acid binding activity in molecular functions; (ii) ROS-related enzymes increased under salt stress. Among them, mitochondrial ascorbate peroxidase was further accumulated by ethanol application; (iii) Among proteins related to membrane and cell wall, xyloglucan xyloglucosyl transferase and H^+^-ATPase increased and decreased, respectively, under salt stress; however, they were recovered to control level by ethanol application. These results suggest that soybeans are adversely affected by salt stress and might be restored by ethanol application through the detoxification of ROS and the regulation of cell wall/membrane function.

## Figures and Tables

**Figure 1 biology-13-00861-f001:**
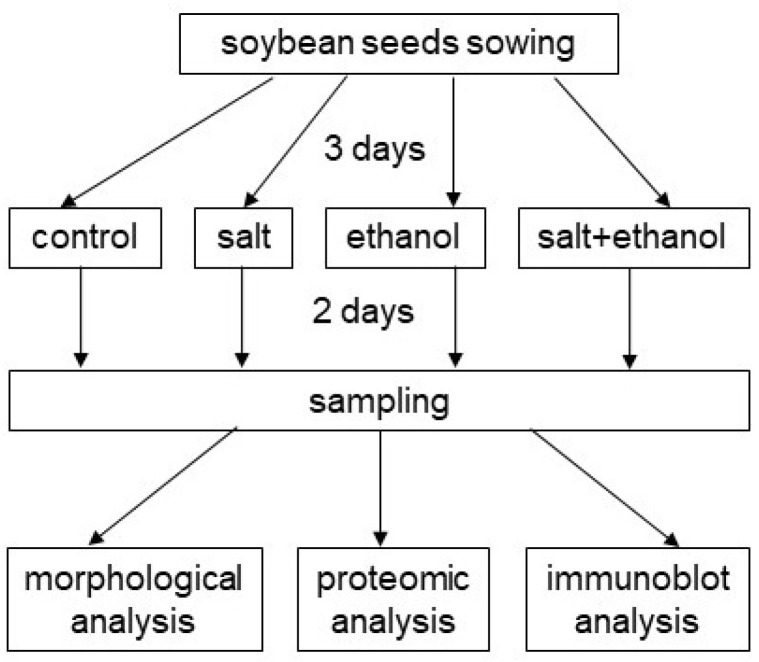
Design of experiments to elucidate the function of ethanol on soybeans under salt stress. Soybean seedlings treated with salt and ethanol were subjected to morphological parameters, and their proteins were further analyzed by proteomics. Proteins identified by proteomics were validated by immunoblots. For all experiments, 3 independent biological replicates were performed.

**Figure 2 biology-13-00861-f002:**
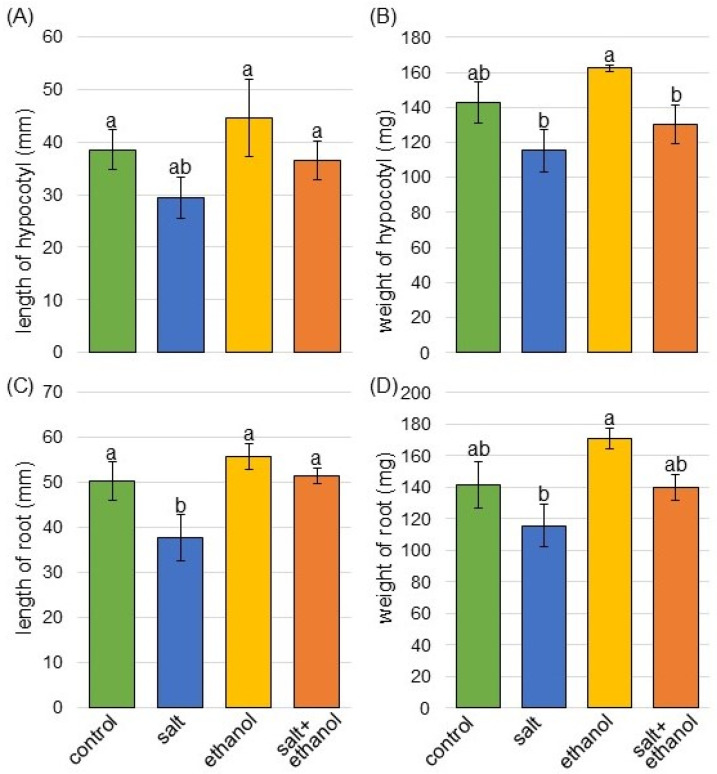
Effect of ethanol on soybean morphology under salt stress. Three-day-old soybean seedlings were treated with or without 300 mM of ethanol as well as with or without 150 mM of NaCl for 2 days. At 5 days after sowing, the hypocotyl length (**A**), hypocotyl-fresh weight (**B**), taproot length (**C**), and total-root fresh weight (**D**) were measured as morphological parameters. Data are presented as mean ± SD from 3 independent biological replicates. The means of points with different letters are significantly different according to a one-way ANOVA followed by Tukey’s multiple comparison test (*p* < 0.05).

**Figure 3 biology-13-00861-f003:**
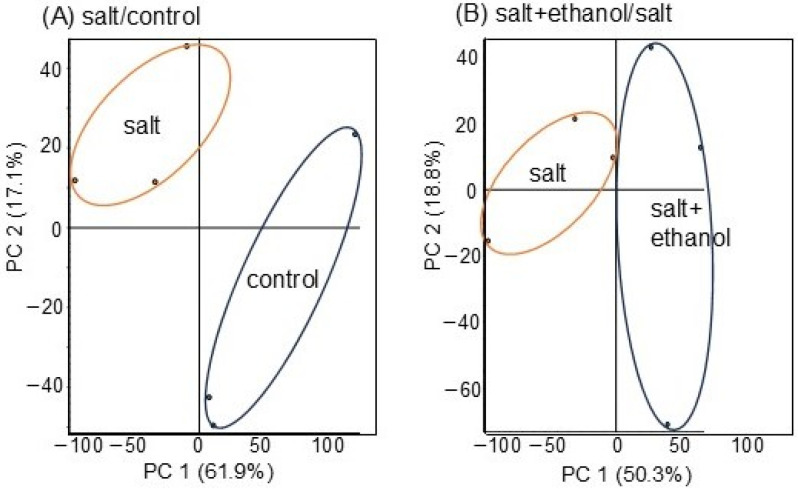
Summary of soybean-proteomic data based on principal component analysis. Three-day-old soybean seedlings were treated with or without 300 mM of ethanol as well as with or without 150 mM of NaCl for 2 days. Soybean roots were collected for protein extraction. Proteomic analysis was conducted in 3 independent biological replicates for each treatment. Principal component analysis was performed in Proteome Discoverer using proteins from 6 different kinds of samples from salt/control (**A**) and salt + ethanol/salt (**B**).

**Figure 4 biology-13-00861-f004:**
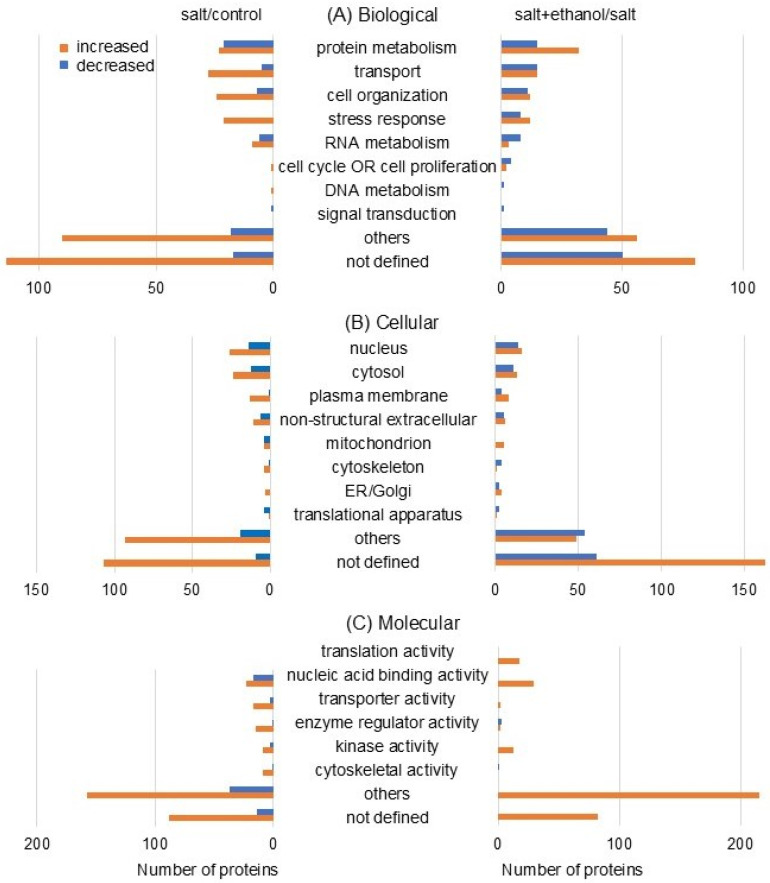
Functional classification of proteins with altered accumulation in the roots of soybeans treated with ethanol under salt stress. Sample collection and experimental methods were the same as in Figure 3. After proteomics, gene-ontology analysis was used to determine functional categories: biological process (**A**), cellular component (**B**), and molecular function (**C**) (Appendix A). The orange and blue columns indicate the number of increased and decreased proteins, respectively.

**Figure 5 biology-13-00861-f005:**
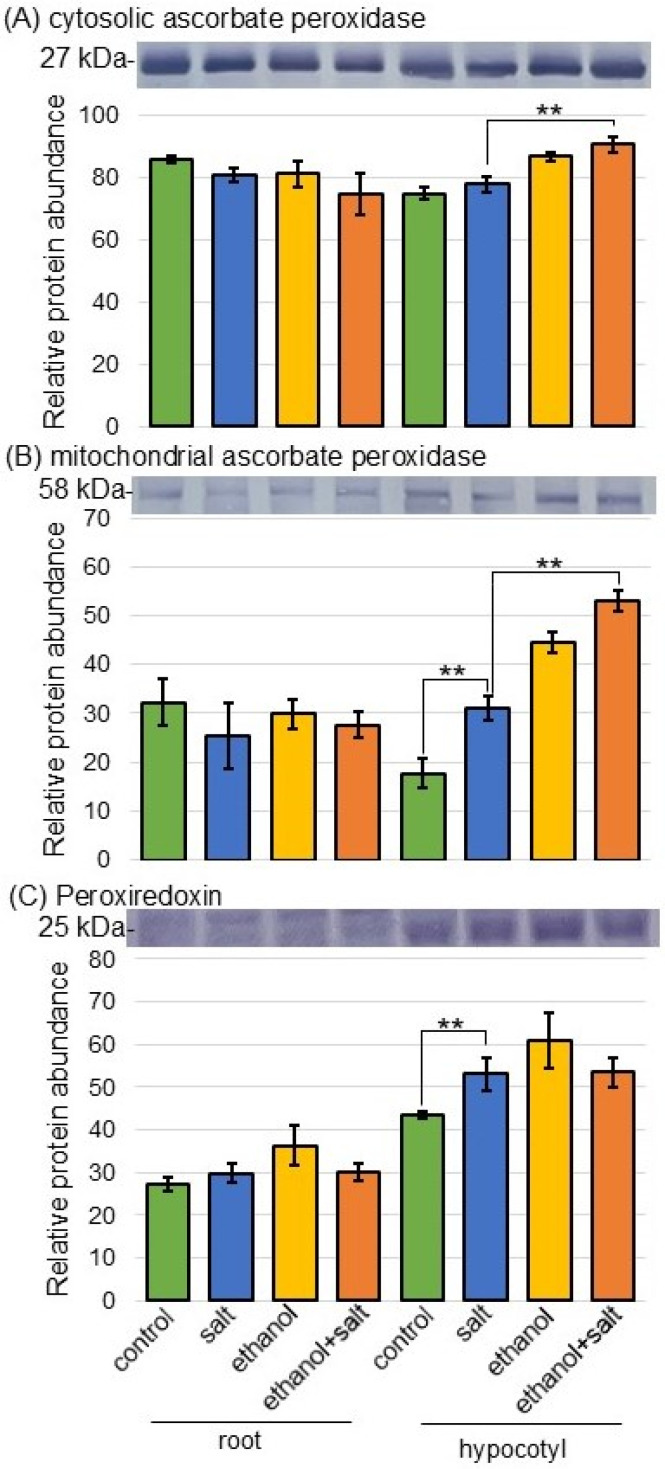
Immunoblot analysis of selected proteins from ROS scavenging enzymes in soybeans treated with ethanol under salt stress. Proteins of soybean roots and hypocotyls were extracted and separated using SDS-polyacrylamide gel electrophoresis. A staining pattern with Coomassie-brilliant blue was used as a loading control (Appendix A). After immuno-reaction, the integrated density of bands was calculated using ImageJ software. As primary antibodies, anti-ascorbate peroxidase (**A**,**B**) and peroxiredoxin (**C**) antibodies were used. Data are presented as the mean ± SD from 3 independent biological replicates (Appendix A). The means of points with star marks are significantly different according to a Student’s *t*-test between 2 groups (**, *p* < 0.01).

**Figure 6 biology-13-00861-f006:**
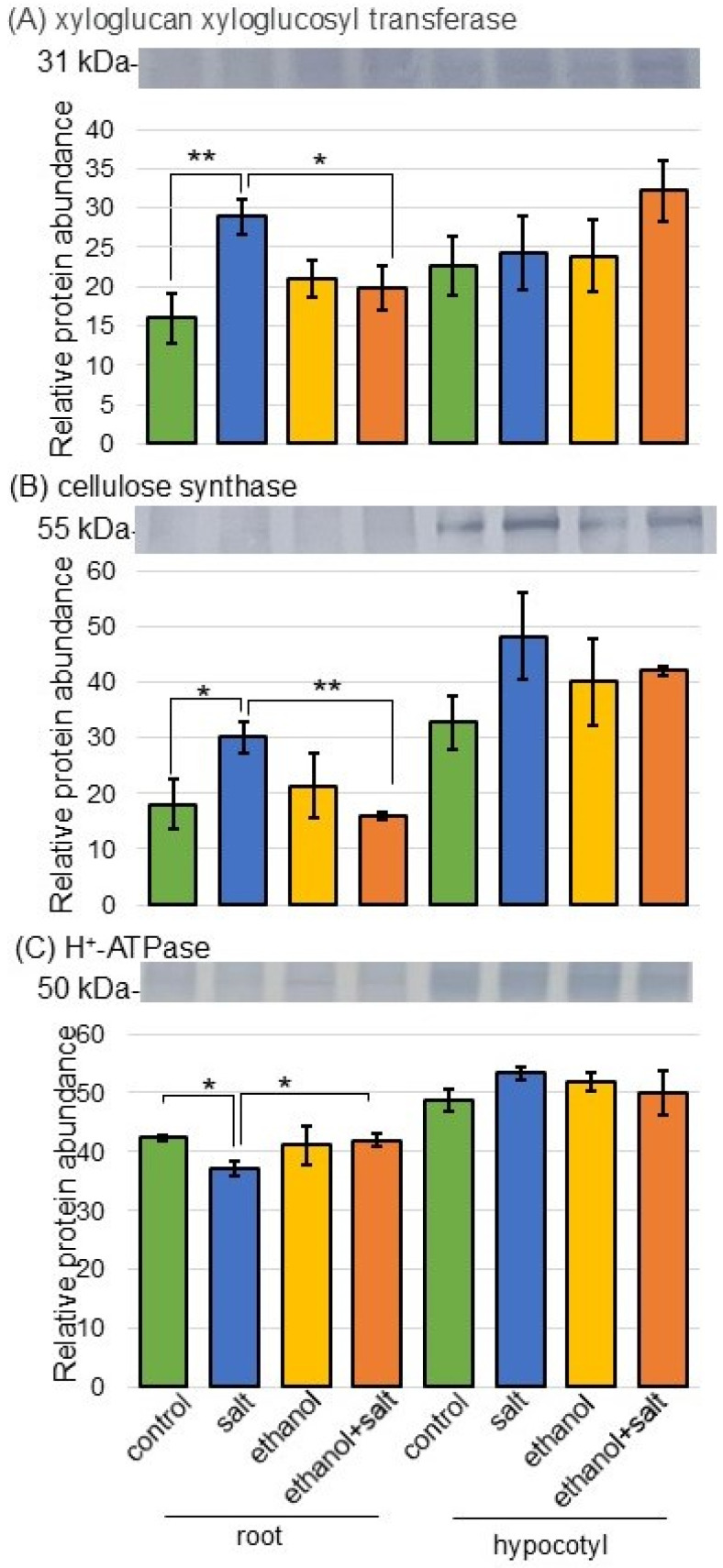
Immunoblot analysis of selected proteins related to the cell wall and membrane in soybeans treated with ethanol under salt stress. Sample collection and experimental methods were the same as in Figure 5. As primary antibodies, anti-xyloglucan xyloglucosyl transferase (**A**), cellulose synthase (**B**), and H^+^-ATPase (**C**) antibodies were used. Data are presented as mean ± SD from 3 independent biological replicates (Appendix A). Statistical analysis is the same as in Figure 5 (*, *p* < 0.05; **, *p* < 0.01).

## Data Availability

For MS data, RAW data, peak lists, and result files have been deposited in the ProteomeXchange Consortium [49] via the jPOST [50] partner repository under data-set identifiers PXD0404581.

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
