# Peer review of "Proteomic Analysis to Understand the Promotive Effect of Ethanol on Soybean Growth Under Salt Stress"

_biology, 2024, doi:10.3390/biology13110861_

Round 1
Reviewer 1 Report
Comments and Suggestions for Authors
Please find the comments regarding the submitted manuscript below:
Introduction
Line 1 – I would suggest, as customary, to insert (in brackets) also the soybean latin name
Line 45 – 'as well as reducing the ratio of these hormones’ seems redundant since this is result of negative regulation of giberrelic acid and positive mediation of abscisic acid
Line 46 – please change ‘phosphor’ to ‘phospho-‘
Lines 55-67 provide an overview of soybean salt tolerance mechanisms which have been extensively studied. I would recommend that more recent reviews on soybean salt tolerance are also included (cited) here.
Materials and Methods
Line 108 – ‘added 400 µL of methanol’, to which 400 µL of methanol was added or 400 µL of methanol was added or similar
Line 152 – please, also correct for English ‘Quantified protein (10 µg) was added sodium dodecyl sulphate (SDS)-sample buffer’
Results
Lines 175-176 – ‘By applying ethanol, these 3 parameters increased even under salt stress.’ I suppose it would be more appropriate to state that they were restored and comparable to control values (not increased)? The same phrasing was used in the Conclusion section/Summary/Abstract
Figure S1, Figure 2 -Figure 4 – please consider including ethanol and salt concentration in the figure captions
Line 195 – Please change the title to include root – ‘Root protein identification’ or similar
Line 207-209 – ‘This result indicated that root proteins were significantly affected by salt stress, which was mild with ethanol application even under stress (Figure 3).’ Something not quite right in this statement language-wise. Please, rephrase. Similar phrasing was used in the Conclusion/Summary/Abstract section and needs to be checked.
Line 212 – The numbers do not add up to 214 (172+43) – please check
Figure 4. – as seen by this reviewer and this layout, increased proteins are colored orange not red
Lines 252/264-265 – Figure S1 and Figure S2 do not match here – the authors probably refer to Tables S1 and S2
Lines 268-269 – ‘On the other hand, H+-ATPase in root decreased under salt stress but 268 was restored to the control level by ethanol application even stress’ – unclear - needs rephrasing/ language editing
Discussion
Line 299 – I suggest replacing ‘many kinds’ with ‘various kinds’ to avoid overinterpretation of the study results
Line 316 – italic not needed in enzyme name
Line 336-337 - There is some unclarity in the statement that ‘ethanol mitigates the life cycle of ROS’ and is somewhat misleading since ethanol is not directly mitigating the life cycle of ROS (but through elevation of ROS scavenging mechanisms)
Comments on the Quality of English LanguageThe manuscript needs minor language editing
Author Response
Reviewer 1
Please find the comments regarding the submitted manuscript below:
Answer: Thank you for your pertinent comments. We have made the nessessary corrections to all of them.
Introduction
Line 1 – I would suggest, as customary, to insert (in brackets) also the soybean latin name
Answer: Thank you very muh for your suggestion. The Latin name for soybean has been added in brackets as follows: “Soybean (Glycine max) is a globally important as the resource crop of oil and protein, but is sensitive to salinity with yield losses up to 40%”.
Line 45 – 'as well as reducing the ratio of these hormones’ seems redundant since this is result of negative regulation of giberrelic acid and positive mediation of abscisic acid
Answer: As suggeted, the words “as well as reducing the ratio of these hormones” have need removed.
Line 46 – please change ‘phosphor’ to ‘phospho-‘
Answer: Thank you very much for your correction. It has been corrected in red.
Lines 55-67 provide an overview of soybean salt tolerance mechanisms which have been extensively studied. I would recommend that more recent reviews on soybean salt tolerance are also included (cited) here.
Answer: Thank you verymuch for your suggetion. The second paragraph of the Introduction has been updated with recent publications as follows:“ High-sodium ion concentrations in soil increase osmotic pressure, disrupt cellular ion homeostasis, and inhibit water/nutrient absorption, which negatively impacts soybean growth and reduces agricultural yields [Yan et al., 2023]. Salt stress is associated with the accumulation of reactive oxygen species (ROS), which act as a secondary stress factor and induce membrane-lipid peroxidation, leading to degrade the structure of cell-membrane proteins [Feng et al., 2023]. Furthermore, the interactions among plant hormone signaling, mitogen-activated protein kinase signaling, phenylpropanoid biosynthesis, starch/sucrose metabolism, and ribosomal metabolism may play important roles in the response of soybean to salt stress [Cheng et al., 2024]. On the other hand, the application of plant-derived smoke solution improved the soybean growth by alleviating salt stress through the regulation of energy metabolism, protein glycosylation, and cell wall construction [Komatsu et al., 2023]. Moreover, the application of safranal to soybean improved salt tolerance by regulating the cell wall along with controlling ROS scavenging system [Kausar et al., 2024]. These findings indicated that modification of multiple intracellular systems could confer salt-stress tolerance to soybean.“
Materials and Methods
Line 108 – ‘added 400 µL of methanol’, to which 400 µL of methanol was added or 400 µL of methanol was added or similar
Answer: Thank you very much for your correction. This sentence has been corrected as follows: “Quantified proteins (100 µg) were adjusted to a final volume of 100 µL, to which 400 µL of methanol was added and mixed followed by the addition of 100 µL of chloroform and 300 µL of water.”
Line 152 – please, also correct for English ‘Quantified protein (10 µg) was added sodium dodecyl sulphate (SDS)-sample buffer’
Answer: Thank you very much for yourcomment. This sentence has been corrected as follows: “Quantified proteins (10 µg) were added to sodium dodecyl sulphate (SDS)-sample buffer”
Results
Lines 175-176 – ‘By applying ethanol, these 3 parameters increased even under salt stress.’ I suppose it would be more appropriate to state that they were restored and comparable to control values (not increased)? The same phrasing was used in the Conclusion section/Summary/Abstract
Answer: Thank you very much for your correction. This sentence has been corrected in the section of Result, Discussion, Conclusion, Abstract, and Simple Summary in red as follows: “ By applying ethanol, these 3 parameters were restored and comparable to control values even under salt stress. “
Figure S1, Figure 2 -Figure 4 – please consider including ethanol and salt concentration in the figure captions
Answer: As suggetsed, ethanol and salt concentrations have been included in Figure S1, Figure 2, and Figure 3.
Line 195 – Please change the title to include root – ‘Root protein identification’ or similar
Answer: Thank you very much for your correction. It has been corrected as suggested.
Line 207-209 – ‘This result indicated that root proteins were significantly affected by salt stress, which was mild with ethanol application even under stress (Figure 3).’ Something not quite right in this statement language-wise. Please, rephrase. Similar phrasing was used in the Conclusion/Summary/Abstract section and needs to be checked.
Answer: We are sorry for this problem. This sentense has been corrected in the section results as follows:“This result indicated that salt stress caused root proteins to separate into significantly different groups, whereas ethanol application brought the two groups closer together.” And this explanation has been removed from Conclusion, Summary, and Abstract sections.
Line 212 – The numbers do not add up to 214 (172+43) – please check
Answer: Thank you very much for your correction. They have been checked and corrected as follows: “ Of the 214 proteins, 173 and 41 proteins increased and decreased, respectively, under salt-stress compared to the control condition “
Figure 4. – as seen by this reviewer and this layout, increased proteins are colored orange not red
Answer: We apologize for this mistake. It has been corrected to orange from red in Figure 4.
Lines 252/264-265 – Figure S1 and Figure S2 do not match here – the authors probably refer to Tables S1 and S2
Answer: We apologize for this mistake. They have been corrected to Tables S1 and S2 from Figures S1 and S2.
Lines 268-269 – ‘On the other hand, H+-ATPase in root decreased under salt stress but 268 was restored to the control level by ethanol application even stress’ – unclear - needs rephrasing/ language editing
Answer: As su\ggested, this sentence has been rephraged as follows: “On the other hand, the accumulation of H+-ATPase in root was decreased under salt stress, but was recovered to the control level by ethanol application even under stress.”
Discussion
Line 299 – I suggest replacing ‘many kinds’ with ‘various kinds’ to avoid overinterpretation of the study results
Answer: As suggeted, it has been corrected in red.
Line 316 – italic not needed in enzyme name
Answer: This expression was difficult to understand, so It has been revised as follows: ” The gene expression level of ascorbate peroxidases was higher in ethanol-treated Arabidopsis than in untreated control plant under salt stress”
Line 336-337 - There is some unclarity in the statement that ‘ethanol mitigates the life cycle of ROS’ and is somewhat misleading since ethanol is not directly mitigating the life cycle of ROS (but through elevation of ROS scavenging mechanisms)
Answer: Thank you very much for your correction. This sentence has been corrected as follows: “These results together with previous studies suggest that ROS generated by salt stress damages the cell membrane and cell wall, but ethanol may alleviate the effects by enhancing ROS scavenging mechanisms.“
The manuscript needs minor language editing
Answer: We are sorry for this problme. Following the comments, this paper has been revised by an American native English speaker.
Reviewer 2 Report
Comments and Suggestions for Authors
The article titled "Proteomic Analysis to Understand the Promotive Effect of Ethanol on Soybean Growth under Salt Stress" provides significant insights into the protective effects of ethanol on soybean growth under salt stress. However, several areas can be improved or clarified for better scientific communication.
The overall English of the manuscript is not good. Throughout the manuscript, there are grammatical errors, awkward phrasing, and punctuation issues that affect readability. The language is not satisfactory for publication at this stage in this journal.
Scientific names of the species and the names of the genes must be italicized in the manuscript.
The introduction provides extensive background on salt stress in soybean, but the role of ethanol in alleviating stress is less well-supported with references. Additionally, the transition from salt stress to ethanol treatment is abrupt.
Line 95: ºC Please write it in the correct form.
Lines 104, 114, 115: Same as above.
Thi discussion part largely repeats the results and does not explore alternative explanations or potential limitations of the study. It also fails to thoroughly discuss the broader implications of ethanol use in agriculture.
The reference list is somewhat inconsistent in formatting, and several key citations appear outdated or irrelevant to the most recent developments. Update references to include more recent studies on ethanol and stress tolerance in plants. Ensure consistent formatting according to the journal’s guidelines.
Many old references have been found. The authors should update the references in the whole manuscript.
Comments on the Quality of English LanguageThe English of this manuscript should be improved.
Author Response
Reviewer 2
The article titled "Proteomic Analysis to Understand the Promotive Effect of Ethanol on Soybean Growth under Salt Stress" provides significant insights into the protective effects of ethanol on soybean growth under salt stress. However, several areas can be improved or clarified for better scientific communication.
The overall English of the manuscript is not good. Throughout the manuscript, there are grammatical errors, awkward phrasing, and punctuation issues that affect readability. The language is not satisfactory for publication at this stage in this journal.
Answer: We are sorry for this problme. Following the comments, this paper has been revised by an American native English speaker. Additionally, this paper has been revised in accordance with the comments.
Scientific names of the species and the names of the genes must be italicized in the manuscript.
Answer: We are sorry for this mistake. As suggested, scientific names of the species and the names of the genes have been italicized in the manuscript in red.
The introduction provides extensive background on salt stress in soybean, but the role of ethanol in alleviating stress is less well-supported with references. Additionally, the transition from salt stress to ethanol treatment is abrupt.
Answer: Thank you very much for your comments. Introduction section has been revised in red. In particular, paragraphs 2 and 3 have been completely revised to make them more coherent in red.
Line 95: ºC Please write it in the correct form.
Lines 104, 114, 115: Same as above.
Answer: We are sorry this mistake. It has been revised in all parts of this paper.
Thi discussion part largely repeats the results and does not explore alternative explanations or potential limitations of the study. It also fails to thoroughly discuss the broader implications of ethanol use in agriculture.
Answer: Thank you for your feedback. The contents in Discussion section have been improved by minimizing overlap with the Results section. The impacts of ethanol use in agriculture are described in Discussion section in red.
The reference list is somewhat inconsistent in formatting, and several key citations appear outdated or irrelevant to the most recent developments. Update references to include more recent studies on ethanol and stress tolerance in plants. Ensure consistent formatting according to the journal’s guidelines.
Many old references have been found. The authors should update the references in the whole manuscript.
Answer: Thank you very much for your comments. The latest papers have been added in sections other than those regarding experimental materials, methods, and databases. The format of references has been corrected for this journal.
The English of this manuscript should be improved.
Answer: We are sorry for this problem. Following the comments, this paper has been revised by an American native English speaker.
Reviewer 3 Report
Comments and Suggestions for Authors
The authors have prepared a study investigating the effects of salt stress and ethanol on soybeans. The authors observed both plant physiological and proteomic manifestations of salt stress and recovery due to ethanol. They found that ethanol treatment is able to recover some effects of salt stress in exposed plants, and there is a differential response in the proteomic profile of each treatment group.
Overall, I see no gross errors in methodology or analysis of the samples in question. The methods used are well established, and frequently utilized in the field.
Though the study is well thought out and executed, there are some methodological details absent from the manuscript that should be included. First, it is stated that plants were ‘treated’ with ethanol or NaCl, but the details of this treatment are absent. Is this foliar application or root applied in aqueous solution? If the latter, what was the volume of water applied to each plant? For the trypsin digestion, the authors state a 1:100 ratio of enzyme to protein, but is this based on mass or molar concentration? For the statistical analysis, was any correction performed for multiple comparisons?
Comments on the Quality of English LanguageThe English grammar is generally adequate in the manuscript, but there are several awkwardly phrased sentences throughout. A review by an English language editor should be able to resolve these problems with ease.
Author Response
Reviewer 3
The authors have prepared a study investigating the effects of salt stress and ethanol on soybeans. The authors observed both plant physiological and proteomic manifestations of salt stress and recovery due to ethanol. They found that ethanol treatment is able to recover some effects of salt stress in exposed plants, and there is a differential response in the proteomic profile of each treatment group.
Overall, I see no gross errors in methodology or analysis of the samples in question. The methods used are well established, and frequently utilized in the field.
Though the study is well thought out and executed, there are some methodological details absent from the manuscript that should be included.
First, it is stated that plants were ‘treated’ with ethanol or NaCl, but the details of this treatment are absent. Is this foliar application or root applied in aqueous solution? If the latter, what was the volume of water applied to each plant?
Answer: Thank you very much for your suggestion. In this experiment, ethanol and NaCl were added as a solution to the sand and absorbed through the roots. The processing method is described in the section of “2.1. Plant Material and Treatment” as follows: “Soybean (Glycine max L. cultivar Enrei) seeds were sown in quarts sand in a nursery case. Three-day-old seedlings were treated with or without 300 mM ethanol (Nacalai Tesque, Kyoto, Japan) and with or without 150 mM NaCl (Nacalai Tesque) for 2 days. For treatment, ethanol and NaCl were added as 100 mL of solution to the sand and absorbed through the roots.”
For the trypsin digestion, the authors state a 1:100 ratio of enzyme to protein, but is this based on mass or molar concentration?
Answer: Thank you very much for your suggestion. It was this was based mass. The processing method is described in the section of “2.3. Protein Enrichment, Reduction, Alkylation, Digestion, and Desalting” as follows: “Alkylated proteins were digested with trypsin (FUJIFILM Wako Chemical, Osaka, Japan) at enzyme:protein mass ratio of 1:100 for 18 h at 37ºC.”
For the statistical analysis, was any correction performed for multiple comparisons?
Answer: In this study, multiple testing was performed on morphological analysis. Experiments were performed with three biological replicates, with 10 plants per replicate. Multiplex testing was performed using three replicates with averaging 10 plants in each replicate. No correction was made for multiple testing because SPSS 20.0 statistical software was just used.
The English grammar is generally adequate in the manuscript, but there are several awkwardly phrased sentences throughout. A review by an English language editor should be able to resolve these problems with ease.
Answer: We are sorry for this problem. Following the comments, this paper has been revised by an American native English speaker.
Round 2
Reviewer 2 Report
Comments and Suggestions for Authors
The authors have incorporated all the suggested changes. This manuscript can be accepted for the publication.
Comments on the Quality of English LanguageMinor editing of English language required.
Author Response
Minor editing of English language required.
Answer: As suggested, this manuscript has been carefully revised.